# PDX models recapitulate the genetic and epigenetic landscape of pediatric T-cell leukemia

Paulina Richter-Pechańska[1,2,†], Joachim B Kunz[1,2,3,†], Beat Bornhauser[4,†], Caroline von Knebel Doeberitz[1,2,†], Tobias Rausch[2,5], Büşra Erarslan-Uysal[1,2], Yassen Assenov[6], Viktoras Frismantas[4], Blerim Marovca[4], Sebastian M Waszak[5], Martin Zimmermann[7], Julia Seemann[1,2], Margit Happich[1,2], Martin Stanulla[7], Martin Schrappe[8], Gunnar Cario[8], Gabriele Escherich[9], Kseniya Bakharevich[9], Renate Kirschner-Schwabe[10], Cornelia Eckert[10], Martina U Muckenthaler[1,2], Jan O Korbel[2,5] , Jean-Pierre Bourquin[4] & Andreas E Kulozik[1,2,*]

## Abstract

We compared 24 primary pediatric T-cell acute lymphoblastic leukemias (T-ALL) collected at the time of initial diagnosis and relapse from 12 patients and 24 matched patient-derived xenografts (PDXs). DNA methylation profile was preserved in PDX mice in 97.5% of the promoters ($\rho = 0.99$). Similarly, the genome-wide chromatin accessibility (ATAC-Seq) was preserved remarkably well ($\rho = 0.96$). Interestingly, both the ATAC regions, which showed a significant decrease in accessibility in PDXs and the regions hyper-methylated in PDXs, were associated with immune response, which might reflect the immune deficiency of the mice and potentially the incomplete interaction between murine cytokines and human receptors. The longitudinal approach of this study allowed an observation that samples collected from patients who developed a type 1 relapse (clonal mutations maintained at relapse) preserved their genomic composition; whereas in patients who developed a type 2 relapse (subset of clonal mutations lost at relapse), the preservation of the leukemia's composition was more variable. In sum, this study underlines the remarkable genomic stability, and for the first time documents the preservation of the epigenomic landscape in T-ALL-derived PDX models.

**Keywords** ATAC-Seq; PDX stability; T-ALL; T-cell leukemia
**Subject Categories** Cancer; Chromatin, Epigenetics, Genomics & Functional Genomics; Haematology

## Introduction

Despite recent improvements in the treatment of pediatric T-ALL (Hunger & Mullighan, 2015), this type of leukemia still represents a major clinical challenge, because relapsed T-ALL face a dismal prognosis inferior to this of B-cell leukemia (Nguyen *et al*, 2008; Van Vlierberghe *et al*, 2008; Dores *et al*, 2012; Locatelli *et al*, 2012; Girardi *et al*, 2017). The development of novel treatment strategies requires a comprehensive understanding of disease biology and valid preclinical models. Despite important limitations of patient-derived xenografts (PDXs) such as clonal selection during propagation in immunodeficient mice (Hidalgo *et al*, 2014), PDX models are indispensable to study biology of leukemia *in vivo* and to test novel molecular-targeted therapeutics (Hidalgo *et al*, 2011; Frismantas *et al*, 2017). Further, the speed of engraftment has been shown to correlate with prognosis, thus highlighting the value of xenotransplanted cells to reflect important clinical variables (Lock *et al*, 2002; Frismantas *et al*, 2017).

It has been shown that PDXs of precursor B-cell ALL, acute myeloid leukemia (AML), and T-ALL retain their leukemogenic profile regarding immunophenotype, chromosomal aberrations,

---

1 Department of Pediatric Oncology, Hematology, and Immunology, University of Heidelberg, and Hopp Children's Cancer Center at NCT Heidelberg, Heidelberg, Germany
2 Molecular Medicine Partnership Unit (MMPU), European Molecular Biology Laboratory (EMBL), University of Heidelberg, Heidelberg, Germany
3 German Consortium for Translational Cancer Research (DKTK), Heidelberg, Germany
4 Division of Pediatric Oncology, University Children's Hospital, Zürich, Switzerland
5 European Molecular Biology Laboratory (EMBL), Heidelberg, Germany
6 Division of Epigenomics and Cancer Risk Factors, German Cancer Research Center (DKFZ), Heidelberg, Germany
7 Department of Pediatric Hematology and Oncology, Hannover Medical School, Hannover, Germany
8 Department of Pediatrics, University Hospital Schleswig-Holstein, Kiel, Germany
9 Clinic of Pediatric Hematology and Oncology, University Medical Center Hamburg-Eppendorf, Hamburg, Germany
10 Department of Pediatric Oncology/Hematology, Charité Universitätsmedizin Berlin, Berlin, Germany
*Corresponding author. Tel: +49 6221 56 4500/+49 6221 56 4555; E-mail: andreas.kulozik@med.uni-heidelberg.de
†These authors contributed equally to this work

transcriptome, and minimal residual disease (MRD) marker expression (Woiterski *et al*, 2013; Wang *et al*, 2017). Cellular barcoding has been used to investigate the clonal evolution of precursor B-cell leukemia cells in PDXs, demonstrating that several individual clones can engraft and expand independently (Belderbos *et al*, 2017). In T-ALL, primary and PDX cells were reported to be generally stable at the genomic level (Wang *et al*, 2017), although the PDX material more closely resembled the relapsed leukemias rather than the pattern observed in initial disease, which suggested that the PDX maneuver selects for subclones that may later give rise to the relapse (Clappier *et al*, 2011). Extensive genomic studies of BCP-ALL showed that PDXs mirror the clonal composition of the original leukemia samples, including from MRD samples, with the exception of losses and gains of alternations in RAS pathway genes, which is also frequently observed in relapsed ALL (Fischer *et al*, 2015; Frismantas *et al*, 2017). Moreover, PDX models of ALL served for identification of a rare subpopulation that resembled relapse-inducing cells whose gene expression profile was similar to primary cells isolated from patients at MRD (Ebinger *et al*, 2016).

Whereas PDX models of leukemia are well-characterized with respect to genomic and transcriptomic features, not much is known about the degree of preservation of the epigenetic profiles in PDX models. We have thus performed a detailed multilevel genomic and epigenomic analysis of primary T-ALL and matched primary PDX samples taken at the time of initial diagnosis and at relapse and find a remarkable genomic and in particular epigenomic stability of pediatric T-ALL when propagated as PDX. While most of the variability affects single nucleotide variants (SNV) that occur at low allele frequency, in many cases the PDX model truly reflects even the leukemia's subclonal composition and clonal hierarchy.

## Results

### PDXs largely preserve genetic alterations and the subclonal architecture of patients' leukemia

We transplanted into NOD/SCID/IL2λ-receptor null (NSG) mice a total of 24 matched samples obtained from 12 patients at the time of initial diagnosis and at relapse. These samples were compared to the respective primary material of the patients (summary of the samples and performed analyses to be found in Dataset EV1). SNP fingerprinting showed that all PDX samples matched the original patients' samples (Appendix Fig S1). On the basis of characteristic expression profiles of the samples propagated in PDX, we classified patients into the following subgroups: TAL1/2 ($n = 5$), TLX1/3 ($n = 3$), HOXA ($n = 1$), NKX2-4/5 ($n = 2$), and LMO2 ($n = 1$; Fig 1C). In addition, we confirmed the breakpoints of the fusions in the genomic DNA in six of 12 primary leukemia samples either by Sanger sequencing (Appendix Fig S2; three patients) or by MLPA analysis for SIL-TAL1 fusions (Dataset EV2a; three patients). In all of these cases, the driving event was identified in the primary samples obtained at initial disease and relapse, and also in the corresponding PDX sample.

We analyzed the genomic stability of the PDXs by comparing SNVs/InDels (Fig 1A and B) and CNA patterns (Fig 1C). 72% (512) of the total 712 SNVs and InDels that were detected in patients' samples were also detected in the corresponding PDXs (all SNVs/

InDels with corresponding AF are listed in Dataset EV2b). Clonal mutations with allele frequencies (AF) ≥ 30% were almost completely (93%) conserved in the PDX samples, whereas subclonal SNVs with an AF < 30% were more frequently absent in PDX samples (194 out of 371, 52%; Fig 1B). The proportion of SNVs preserved was higher at initial disease (175/213; 82%) compared to relapse (337/499; 68%), although this difference was mainly driven by a single relapse sample from patient P1 where one subclone with 98 mutations with an allele frequency below 20% did not engraft (Appendix Fig S3).

Thirty (86%) of the 35 large CNAs, identified by low-coverage WGS, spanning from several Mb up to entire chromosomes (Appendix Fig S4), could also be detected in the corresponding PDX models (Fig 1C). Independent CNA analysis with multiplex ligation-dependent probe amplification (MLPA), allowing higher sensitivity for detection of known CNAs, was performed in 17 regions, covering 16 T-ALL-related genes (Dataset EV2a). This analysis shows that PDXs retained 96% (80/83) of the deletions and amplifications present in the primary patient samples.

Altogether, 20 out of 24 PDXs preserved all clonal mutations (AF ≥ 30%) detected in the patients' leukemia (Fig 2A). Interestingly, out of the 19 samples with at least one subclone (defined by mutations with an AF < 30%) in the primary sample, in 17 samples at least two clones were detected in the corresponding PDX model. The mean AF of the preserved mutations was slightly higher in PDX models (38%) than in the patients' samples [33%; $P < 0.0001$ (Wilcoxon)], which indicates enrichment of either leukemia cells or of particular subclones in the PDX. Moreover, a Pearson's correlation coefficient of 65% between AF of mutations detected in patients' samples and in PDXs suggests that clonal hierarchy expressed as a proportion of the cells that carry particular SNV/InDel was well preserved in the PDXs (Fig 2B). An overview of the fraction of preserved mutations for each of the samples is shown in Dataset EV3.

### Primary T-ALL cells can engraft in defined modes that depend on the subclonal architecture

We have next focused on the comparison of engraftment patterns between patients who later developed either a type 1 (all clonal mutations detected at the time of initial disease maintained at relapse) or a type 2 (a subset of clonal mutations detected at initial disease lost at relapse) relapse (Kunz *et al*, 2015) and hypothesized that the clonal complexity of the initial T-ALL in patients who later developed a type 2 relapse may be less well preserved in the PDX than in patients who later developed a type 1 relapse. In these groups of patients, we were particularly interested in whether the clonal composition (SNVs/InDels with AF ≥ 30%) and the clonal architecture (selection/eradication of specific subclones/clones) are preserved. In all of the patients who later developed a type 1 relapse, the clonal SNVs found in the primary initial sample could also be detected with an allele frequency of ≥ 30% in the corresponding PDXs [patients: mean AF 46.5% (SD: 12.6%); PDXs: mean AF 46.3% (SD: 13.8%)].

By contrast, in seven patients who later developed a type 2 relapse, the engraftment pattern of the samples obtained at the time of initial disease was more variable. The genomic complexity of these samples was only maintained in two of seven corresponding

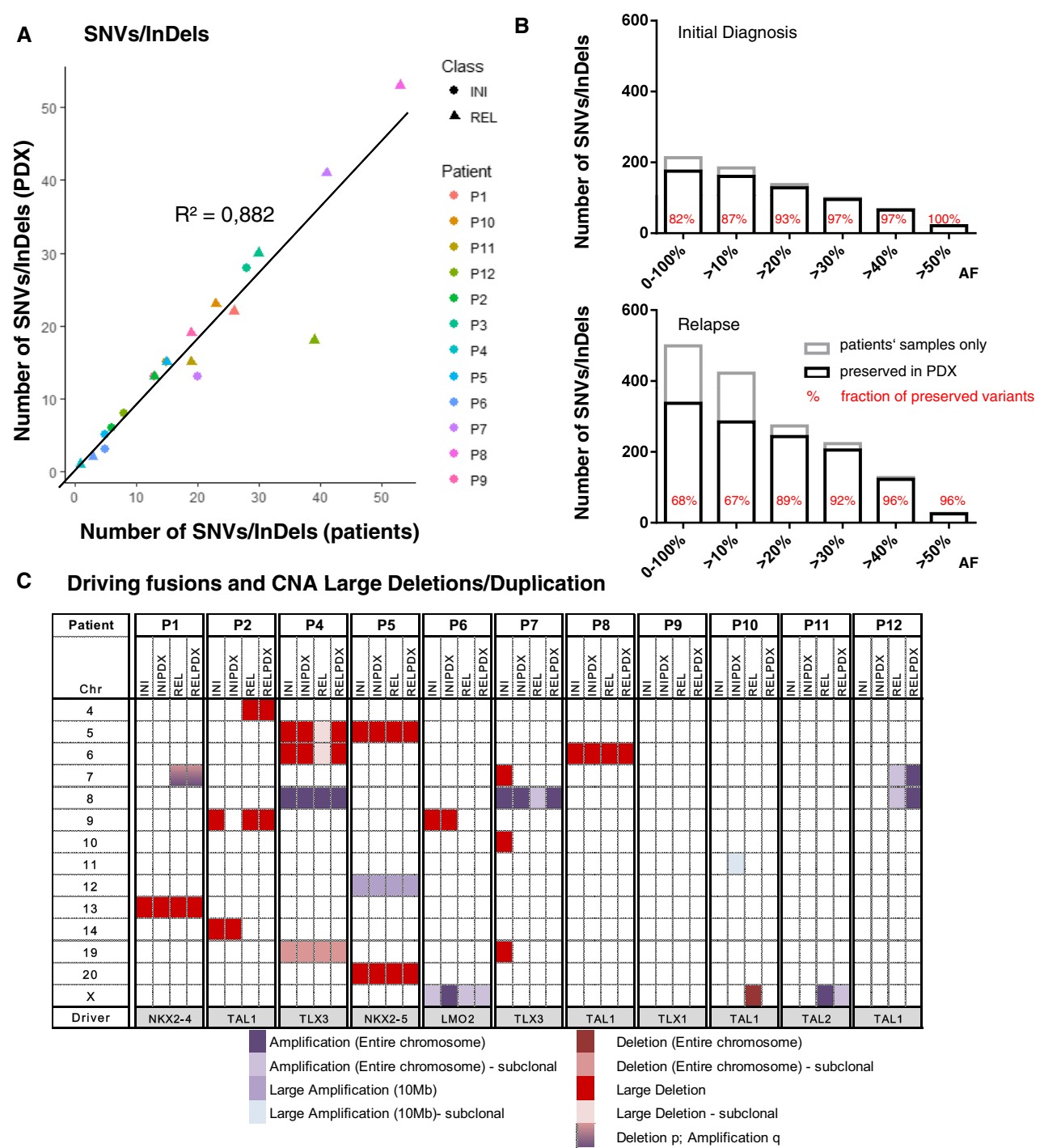

**Figure 1. PDX models recapitulate genomic features of the corresponding patients' leukemias.**

A   Total number of SNVs/InDels (allele frequency ≥ 20%) detected in primary samples (x-axis) plotted against the number of SNVs/InDels detected in corresponding PDX samples (y-axis); $R^2$—coefficient of determination.

B   Number of preserved SNVs/InDels (black bars) and those that were not detected in PDX models (gray bars) in relation to their allele frequencies (AF); %—fraction of preserved mutations.

C   Large (> 1 Mb) CNA (amplifications, shades of violet; deletions, shades of red) identified by low-coverage whole-genome sequencing in each of the patients and corresponding PDXs.

PDXs (P1 and P8; Fig 2), whereas in five of seven PDXs, the clonal composition or architecture was remodeled. Of these five models, three (P2, P10, and P12) preserved all clonal mutations (AF > 30%). However, the actual AF varied between the primary and the PDX samples likely reflecting clonal selection (Fig 2: blue; for details, see Dataset EV2b). In two of these PDX models (P7 and

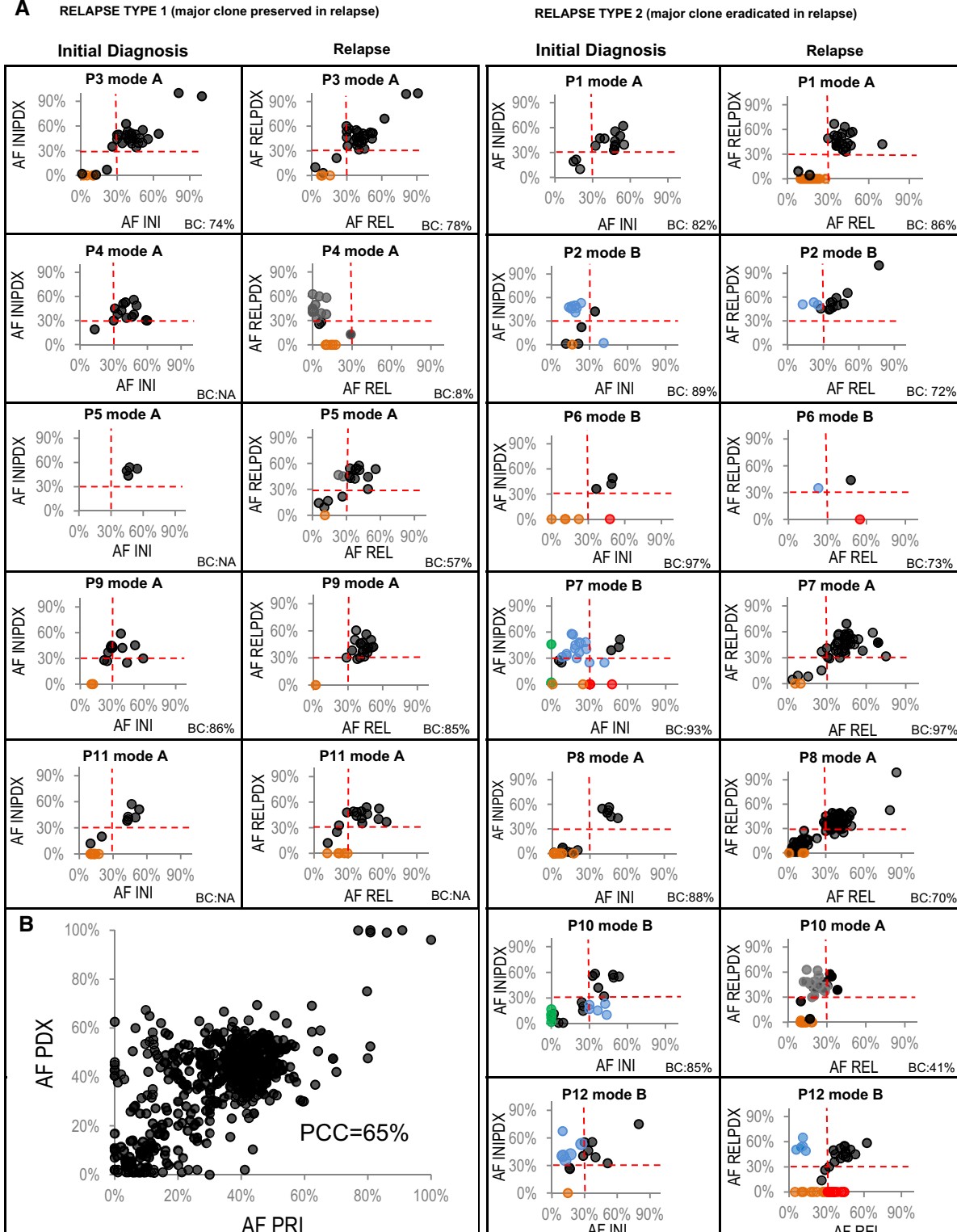

**Figure 2. Engraftment mode (A—concordant; B—discordant) correlates with the type of relapse.**

A  Allele frequencies (AF) of mutations detected in primary (PRI) patients' samples (x-axis) at initial diagnosis (INI) and relapse (REL) plotted against AF of corresponding mutations detected in matched PDX models (y-axis) for type 1 relapse (left) and type 2 relapse patients (right); black—mutations preserved between primary and PDX; blue—mutations selected for/against in PDX models; green—relapse-specific mutations selected for in INIPDX, absent or detected with a very low AF in primary patients' diagnosis sample; orange—subclonal mutations lost in PDX; red—clonal mutations lost in PDX; gray—mutations, in which AF was affected by low blast content (BC) at relapse.

B  Allele frequencies of all mutations preserved between primary patients' samples (x-axis) and PDX samples (y-axis); PCC—Pearson's correlation coefficient.

P10), we detected mutations that were acquired at the time of relapse in the original sample (Fig 2: green). This finding indicates that the clone responsible for the relapse was already present at a subclonal level at initial diagnosis but could not be detected at an average read depth of 50× at that time. Finally, two of these models (P6 and P7) did not preserve some alterations (SNVs and large deletions of chr7&10) found in the primary initial leukemia. These mutations were not detected at relapse either, which may indicate that they represent passenger mutations. Alternatively, these mutations may have been present in subclones that were selected for in the progression of the major population prior to diagnosis, but were disfavored by both treatment and engraftment, leading to selection for an "earlier" divergent subclone.

On the other hand, the genomic composition of relapse samples was much better maintained in corresponding models of all five patients who had developed a type 1 relapse and in four of seven patients with a type 2 relapse. Altogether, in 10 out of 12 PDXs of relapse, all clonal mutations that were found in patients' samples were also detected in the corresponding PDX (Fig 2). Notably, in cases with low blast content in the primary sample (P4, P5, and P10), the PDX enriched for leukemia cells, thus facilitating a representative analysis (Fig 2). Finally, in two PDXs of relapsed leukemia, one of two clones detected in the patients' samples was selected for in the PDX, while the other one was lost (P6 and P12). The absence of the variants in the matched PDX models was confirmed by Sanger sequencing for a subset of five mutations in *STAT5B* (N642H), *DNM2* (R199Q), *FOXO3* (R211Q), *PIK3CA* (R1023Q), and *MAP1B* (T908K; Appendix Fig S5A).

## PDX-specific mutations

In 10 of 24 PDXs, we detected a total of 38 SNVs and three InDels with an AF > 10% that were not detected in the patients' samples, neither at initial diagnosis nor at relapse (Dataset EV2c). None of these genes was recurrently mutated. Four of the mutations were previously reported in the COSMIC database, one of which was a known activating variant of *NOTCH1* (L1585P; Breit *et al*, 2006), which is concordant with a previous report showing that PDX often contain additional mutations in established human oncogenes and/or tumor suppressor genes (Clappier *et al*, 2011). However, allele frequencies of the PDX-specific variants co-occurring in the same model were constant, which indicates that the variants were carried by the same clone. Therefore, we suggest that these variants tend not to be truly acquired during the xenotransplantation maneuver, but were likely present in the original patient's sample, but at an AF below the detection limit of our analysis and were likely selected in the PDX. We did not observe any CNA to be acquired during the process of engraftment. The absence of mutations in *NOTCH1*, *SGCE*, *WDR88*, *MAST3*, *PCDH15*, and *COL6A3* in the primary leukemias (P1INI, P5INI, P10INI, P10REL, and P12REL) was validated by Sanger sequencing as shown in Appendix Fig S5B.

## DNA methylation is stably propagated in PDX models

We used Illumina 450k and 850k arrays to analyze DNA methylation in the 22 of 24 paired samples for which DNA was available

and observed that the use of the two different types of arrays did not introduce any bias (Fig EV1). Compared to patients' samples, the global promoter methylation profiles in the PDXs were almost identical (Appendix Fig S6). Similarly, unsupervised hierarchical clustering based on the average degree of methylation of the 1,000 most variable promoters suggested a high degree of relationship between the patients' leukemias and the corresponding PDX models and indicates that the data are not confounded by murine contamination (Fig 3A). The only exception, the relapse sample of patient P4, which did not cluster together with the sample obtained at the time of initial disease, could be explained by the low blast content of the relapse sample of only 8% indicating that the PDX maneuver enriched for leukemia cells from a background of normal cells.

The degree of promoter methylation in patients' samples plotted against the degree of promoter methylation in PDX samples confirmed the high concordance between the two groups [PCC (Pearson's correlation coefficient) = 0.9938; Fig 3B], although a slight trend for increased methylation levels of the promoters could be observed in the PDX models [$P = 0.1531$ (Wilcoxon rank-sum test)]. Altogether, we found 2.5% of all the analyzed promoters to be recurrently differentially methylated (Dataset EV4), which is in agreement with previous reports (Guilhamon *et al*, 2014; Tomar *et al*, 2016). Moreover, we found the expected negative correlation between the blast content and the total number of differentially methylated promoters (β value difference of at least 0.2) of −88% (PCC). Thus, the higher proportion of differences in relapse samples (Fig EV2) can largely be explained by a lower blast content (mean: 69%; SD: 27%) in comparison with samples collected at initial diagnosis (mean: 87%; SD: 7%). The number of differences detected at the epigenetic level between primary leukemia and the matched PDX for each of the analyzed samples is listed in Dataset EV3. Altogether, there were 69 promoters that were recurrently hypomethylated in PDX models and 484 promoters that were recurrently hypermethylated. As the 69 promoters in which we observe reduced methylation levels in PDX are most likely a result of the low blast content in some of the primary samples (Fig EV2), we focused on the 484 promoters hypermethylated in the PDX models. Publicly available expression data from 264 T-ALL patients (Liu *et al*, 2017) were available for 408 of these genes. Of the 408 genes, only 12 (3%) had an average expression of at least five FPKM and additional 21 genes (5%) had a variable expression (SD ≥ 3, expressed with at least five FPKM in at least three patients). In comparison, 43% (8,412/19,464) of the entire transcriptome were categorized as either expressed or variably expressed. Three hundred and seventy-five of the 408 genes hypermethylated in PDX were not expressed at all or expressed at a very low level (average FPKM ≤ 2), indicating that the vast majority of hypermethylated genes are transcriptionally silent in T-ALL (Fig EV3).

Functional enrichment analysis of the remaining 33 genes (listed in Dataset EV4) that were (variably) expressed yielded a term GO:0060761: "negative regulation of response to cytokine stimulus" as most significantly enriched (FDR = 7.33e-03). Altogether, these data suggest that the majority of the genes (92%) that were hypermethylated in PDX were already silenced prior to the propagation in mice, which is in accordance with previously published observations that methylation might serve not to actually repress previously transcribed genes, but maintains their continued silencing in somatic tissues that might otherwise be permissive to these genes (Klutstein

**A**

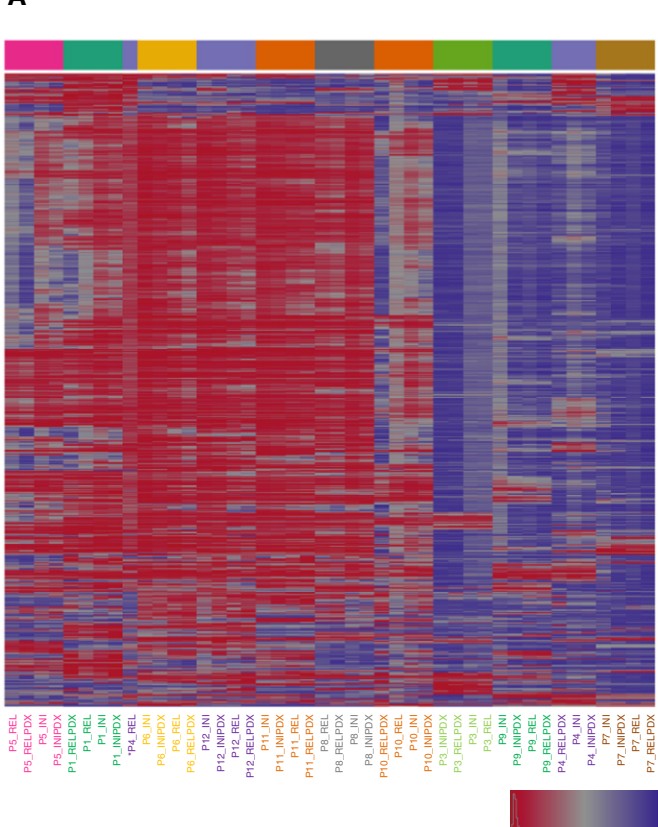

**B**

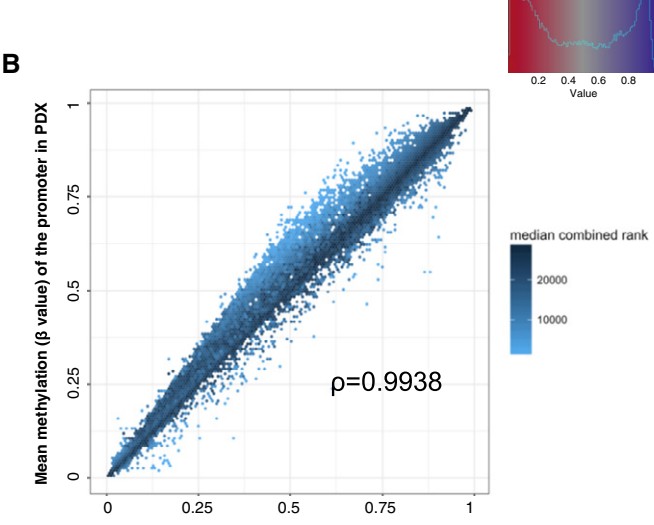

Figure 3. The epigenomic profile reflected by DNA methylation is recapitulated in NSG mice.

A  Unsupervised hierarchical clustering based on the average degree of methylation of the 500 most variable promoters (red—low/violet—high methylation levels); * relapse (REL) sample of patient P4 with blast content of 8%.

B  The degree of the mean promoter methylation in patients' samples plotted against the degree of mean promoter methylation in PDX samples; ρ—Spearman's rank correlation coefficient; median combined ranks across hexagonal bins are shown as a gradient according to the color legend.

*et al*, 2016). In a small proportion of differentially methylated promoters (8%; 0.17% of all the promoters), the increase in the methylation level in PDX models might be a result of the incomplete interaction between murine cytokines and human receptors.

In summary, PDX models exhibit a very stable methylome and all the PDX models reflected the pattern of the primary leukemias. Altogether, 97.5% of promoters are not significantly altered in PDX, while only 2.5% is differentially methylated upon propagation in mice.

**Chromatin accessibility and nucleosome patterns are maintained during engraftment**

We used Assay for Transposase-Accessible Chromatin (ATAC) sequencing (Corces *et al*, 2016) to determine differences in the genome-wide chromatin accessibility landscape in six patients for whom primary and PDX materials were available. For three of the patients, we analyzed biological replicates of PDXs (two different mice transplanted with the same primary leukemia).

Quality control of the libraries (Dataset EV5a) revealed that the fraction of the reads in peaks and the transcription start site (TSS) enrichment score was significantly higher in PDXs than in libraries prepared from primary samples [$P = 0.03$ (Wilcoxon matched-pairs signed rank test) Figs 4A and EV4] indicating that the technical quality of the PDX material is higher than that of the primary leukemia. This difference likely reflects the various pre-analytical factors that affect the primary leukemia samples that have been obtained in the context of multicenter studies. These technical challenges relating to the primary leukemia samples are also reflected by the low quality of the RNA in these samples, which precluded generation of interpretable RNA-Seq datasets.

Analysis of the fraction of shared peaks showed that 91.5–94.5% of ATAC peaks with a height (average number of reads that map into an accessible region called by Macs2) of ≥ 50 are preserved between the primary patient's leukemia and the corresponding model (Fig 4B). Moreover, the average number of reads per peak reached a very high concordance (Fig 4C, mean coefficient of determination of 0.92–0.98). Unsupervised learning by principal component analysis (PCA) clusters samples originating from the same patient in close proximity (Fig 4D) demonstrating that the peak profiles are largely preserved during propagation in mice. Moreover, the stability of the process of engraftment is reflected by particularly the close localization of the biological replicates of PDX models on the PCA plot. Moreover, a comparison with previously published chromatin datasets for the T-ALL cell line DND-41 (Knoechel *et al*, 2014), in which we computed expected values based on the randomly shuffled peaks, shows a high degree of overlap between the ATAC peaks and the active promoters and enhancers detected in histone methylation/acetylation analysis by chromatin immunoprecipitation and sequencing (Fig EV5).

To identify differentially accessible promoter regions between primary and PDX samples, we performed differential peak calling using DESeq2 (Love *et al*, 2014). Out of 77,344 (10,755 TSS and 66,589 non-TSS) peaks called in the analyzed sample pairs, 2,667 (3.4%) showed significantly increased accessibility (hyper-accessible) and 2,887 (3.7%) had significantly decreased accessibility (hypo-accessible) in PDX samples in comparison with the primaries ($P < 0.05$; Fig 4E; Dataset EV5b). Representative ATAC-Seq tracks for known leukemia drivers that are accessible both, in the primary

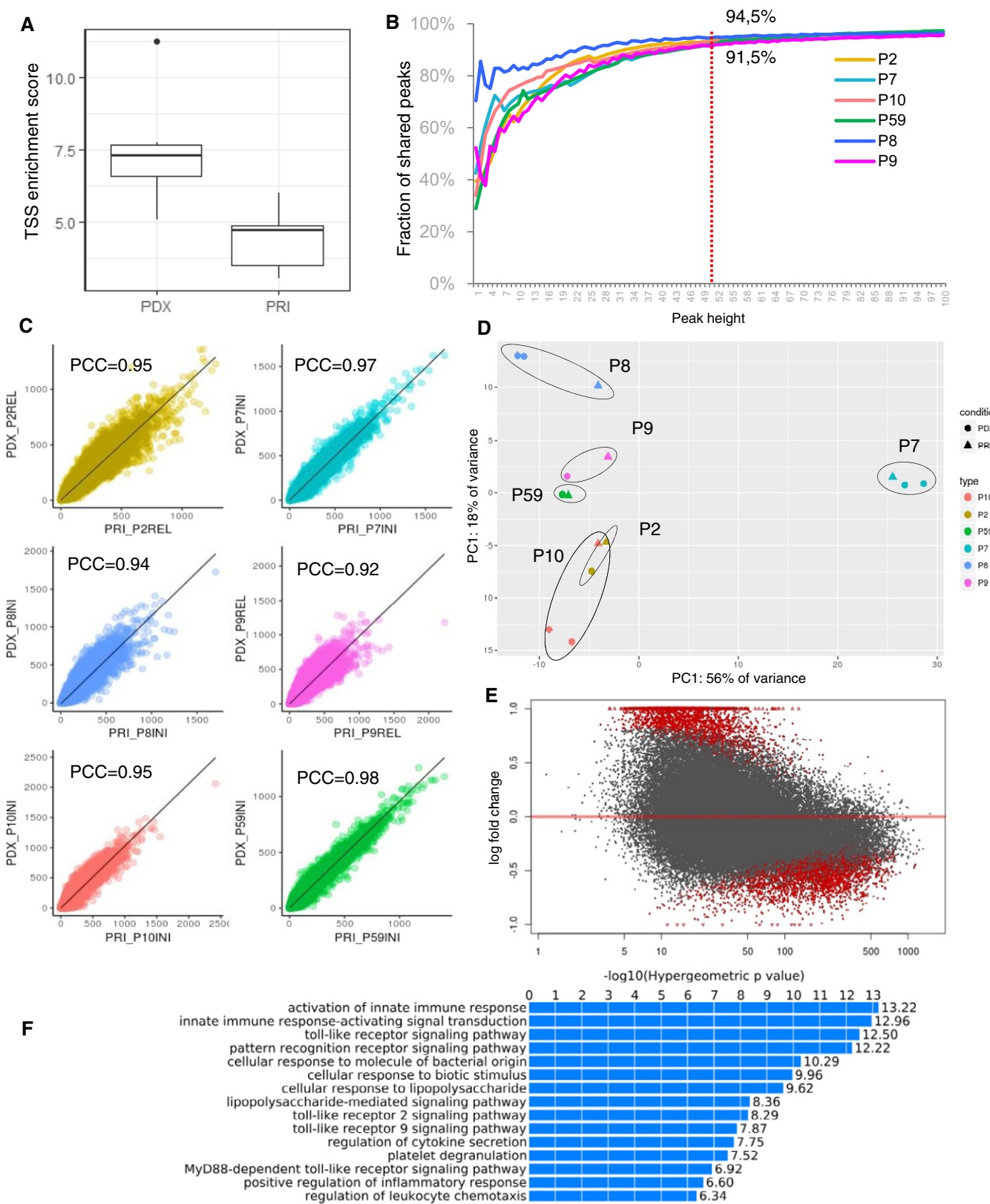

**Figure 4.**

**Figure 4.    Chromatin accessibility profiles of primary leukemias are maintained in PDX.**

A    TSS enrichment score of primary samples (PRI, $n = 65$) and PDXs ($n = 9$). Horizontal lines indicate median, lower and upper limits of each box correspond to the first and third quartiles (the 25th and 75th percentiles) and the lower and upper whiskers extend from min to max.

B    Reads number per peak in primary samples (x-axis) plotted against the reads number in corresponding PDX model; PCC—Pearson's correlation coefficient.

C    Fraction of peaks shared between PDX and corresponding primary leukemia for ATAC peaks with a certain height (x-axis; average number of reads that map into an accessible region called by Macs2).

D    Unsupervised learning by principal component analysis (PCA); circle—primary sample (PRI); triangle—PDX sample; each color corresponds to 1 patient.

E    MA plot of log2 fold changes between primary and PDX samples plotted against the mean of normalized read counts per peak, red indicates differentially accessible regions ($P < 0.05$) as analyzed by DESeq.

F    GO term enrichment of the predicted cis-regulatory regions downregulated in PDX in comparison with primary samples performed by Genomic Regions Enrichment of Annotations Tool (GREAT).

and in the PDX samples, and for the differentially accessible peaks, are shown in Appendix Fig S7.

Functional enrichment analysis was performed separately for the TSS peaks and for the distal (non-TSS) peaks. In both cases, this analysis did not yield any ontology term within the hyper-accessible peaks, whereas the hypo-accessible signature was highly enriched in terms associated with immune/defense responses, cytokine production, and leukocyte activation (Fig 4F). These data suggest a reduced interaction between the immune system and the leukemia in PDX models and possibly the incapability of murine cytokines to stimulate corresponding responses. As a result, in PDX models, we observed induced chromatin condensation of the gene regulatory elements involved in immune function and in the regulation of cytokine production of in comparison with their matched primary leukemias (examples are shown in the Appendix Fig S7).

In sum, however, the overall chromatin accessibility measured by ATAC-Seq is preserved remarkably well in the PDX models.

## Discussion

Patient-derived xenografts have emerged as a useful platform to model cancer biology and to develop new treatment strategies. Previous studies have shown the utility of PDXs in modeling of acute T-cell leukemia and the importance of genomic profiling of the models to ensure concordance with the primary sample of origin (Wang et al, 2017). As the importance of the epigenetic level of gene regulation in cancer cells generally and in leukemia in particular has become increasingly apparent (Dawson & Kouzarides, 2012; Peirs et al, 2015), it is a fundamental question in how far the transfer of primary leukemia cells into PDX models affects the epigenetic profile of these cells. It is one of the important new findings reported here that the epigenetic profiles of pediatric T-ALL cells are faithfully reflected in PDX mice. More than 97.5% of the promoters covered by the arrays to analyze DNA methylation are stably preserved when xenotransplanted into NSG mice. Similarly, ATAC sequencing revealed a large conservation of the genome-wide chromatin accessibility profiles in PDXs. Moreover, the analysis of the biological replicates in which the same leukemia was transplanted into two different mice shows that chromatin accessibility profiles are stably propagated during engraftment. Interestingly, however, a 3.7% fraction of peaks with significantly decreased accessibility in PDX mice as well as 33 promoters recurrently hypermethylated in PDXs when compared to the primary samples are highly enriched for immune function categories, which suggests that the immunodeficient background of the mice has a noticeable effect on the chromatin landscape and methylation patterns of the leukemia cells.

As a second important new finding, we show in a longitudinal analysis of the sample pairs obtained from the same patient at the time of initial disease and at relapse that the mode of engraftment of the sample obtained at initial disease is correlated to the type of relapse (Kunz et al, 2015). Samples of patients who later develop a type 1 relapse are more stable in the PDX mice than those that have been obtained from patients with a type 2 relapse, which reflects the more complex subclonal architecture of the high-risk leukemia giving rise to a type 2 relapse. In analogy to the two modes of type 1 relapse and type 2 relapse, we propose two modes of engraftment. In the first mode that we refer to as mode A, both, the clonal composition and the clonal architecture, are captured in the corresponding PDX. In the second mode that we call mode B, the composition of the major clone including its characteristic SNVs and CNAs and/or the clonal architecture is remodeled in the PDX by clonal selection (Fig 5). Primary samples collected at the time of initial disease of patients who later develop a type 1 relapse (clonal mutations detected at the time of initial disease are maintained at relapse) tend to preserve the genomic composition and the genomic architecture in the PDX. In contrast, only in two of seven patients with type 2 relapse, the complexity of the primary sample collected at initial disease is maintained during the xenotransplantation maneuver (mode A; Fig 5), whereas in the five remaining patients, the clonal composition or/ and the clonal architecture is remodeled (mode B). In two of the patients, a subclone that later gives rise to the relapse is selected in the xenotransplanted cells from the initial presentation (Fig 2). This finding supports a previous report indicating that the genomic composition of PDX might more closely resemble relapse samples than samples obtained at the time of initial diagnosis (Clappier et al, 2011). However, in none of the cases, the relapsing subclone outgrew a clone specific for the patients' initial diagnosis. Moreover, in no case, we observed that the PDX from the time of initial diagnosis already displayed the full complexity of relapse as all relapsed leukemias had gained additional mutations (Appendix Fig S8), indicating that clonal selection is only one of the mechanisms shaping relapse. Despite many acquired subclonal mutations, which result in a higher genomic heterogeneity, samples collected at the time of relapse tend to better maintain their genomic composition when engrafted in NSG mice when compared to samples collected at initial diagnosis. We suggest that the propensity of relapse samples to preserve their clonal architecture in the PDX is a result of the clonal selection which the leukemia has undergone on the way to relapse in the patient.

Previous studies addressed the question of clonal composition and selection during engraftment of ALL either by analysis in a bulk cell population (Anderson et al, 2011; Clappier et al, 2011; Ben-David et al, 2017) or at single-cell resolution (Belderbos et al, 2017) and showed that leukemia is driven through a dynamic pattern with

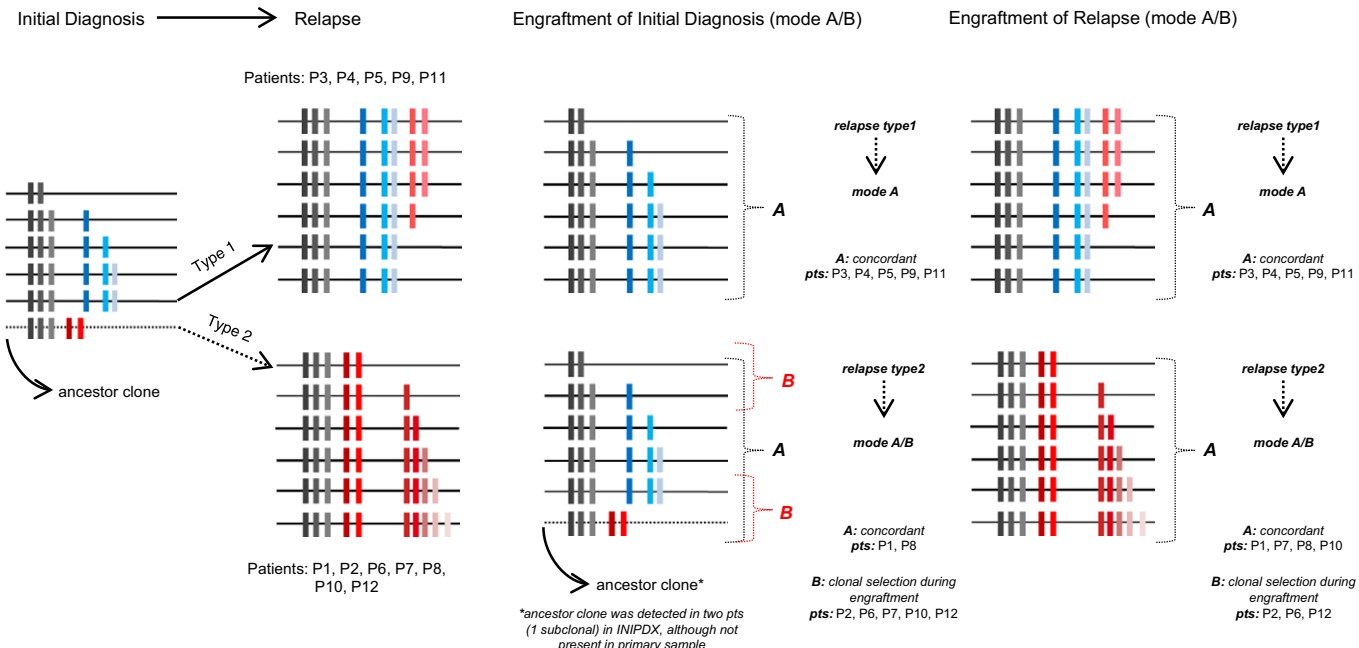

**Figure 5. Schematic view of the correlation between the modes of engraftment with the type of relapse.**

In engraftment mode A, both the clonal composition (all clonal mutations with AF > 30%) and the clonal architecture are captured in the corresponding PDX. In mode B, the composition of the major clone including its characteristic SNVs and CNAs and/or the clonal architecture is remodeled in the PDX by clonal selection; gray—core mutations common for both initial diagnosis and relapse; blue—mutations specific to initial diagnosis; red—relapse-specific mutations.

variable branching trajectories instead of sequential linear processes (Anderson *et al*, 2011). It was demonstrated that engraftment is a stochastic process in the primary passage and a deterministic clone-size-based process in the secondary and tertiary passages with several clones having leukemia propagating capacities (Belderbos *et al*, 2017). Moreover, it has recently been shown that the preselection of minor subclones during PDX passaging results in a rapid accumulation of copy number alterations and aneuploidy, which differ from those acquired during tumor evolution in patients (Ben-David *et al*, 2017). These findings together with an ongoing reduction in the number of clones in serial xenografts (Belderbos *et al*, 2017) argue against the usage of secondary and tertiary models and suggest that primary passages, such as used in our study, faithfully reflect the clonal diversity of human T-ALL.

In conclusion, pediatric T-ALL cells largely maintain their DNA methylation pattern, their chromatin architecture, and most of their genomic characteristics when xenotransplanted into primary NSG mice. However, it must be noted that the absence of a normal immune system and the reduced ability of murine cytokines and chemokines to replace their human counterparts (Francis *et al*, 2016) result in an understimulation of immune pathways. The data presented in this report indicate these effects to reduce the chromatin accessibility in a small fraction of gene loci, as reflected by ~3.7% of the analyzed ATAC peaks, and to induce methylation in the small fraction of the gene promoters (0.17%).

Moreover, some of the complexity of the genomic characteristics and the subclonal architecture may be remodeled in the course of engraftment. Our results indicate that such a remodeling process preferentially affects samples obtained at the time of initial disease and particularly when collected from patients who later develop a

type 2 relapse. Although this mode of engraftment must be considered when interpreting data derived from PDX models, the overall genomic and epigenomic stability of T-ALL following transfer into NSG mouse indicates that primary PDX models are suitable surrogates for the study of disease biology and for the preclinical development of novel treatment strategies.

# Materials and Methods

### Patients' clinical characteristics

The primary cells were obtained from patients recruited in ALL-BFM 2000, ALL-BFM-2009, CoALL97, CoALL03, CoALL09, and ALL-REZ BFM 2002 trials and were selected on the basis of sufficient material being available from the time points of first diagnosis, remission, and relapse. For patients' clinical characteristics, see Dataset EV6.

Clinical trials from which samples were used in this analysis had previously received approval from the relevant institutional review boards or ethics committees. Written informed consent had been obtained from all the patients and the experiments conformed to the principles set out in the WMA Declaration of Helsinki and the Department of Health and Human Services Belmont Report.

### Establishment of the patient-derived xenografts

Patient-derived xenografts (PDX) were generated as described (Schmitz *et al*, 2011) by intrafemoral injection of $1 \times 10^5$ to $5 \times 10^6$ viable primary ALL cells in NSG (NOD.Cg-PrkdscidIl2rgtm1Wjl/SzJ) mice. Transplanted mice were both male and female, aged 5–

8 weeks. Animals were housed in individually ventilated cages with access to food and water *ad libitum*. Leukemia progression was monitored in the peripheral blood by flow cytometry using anti-mCD45, anti-hCD45, anti-hCD19, or anti-hCD7 antibodies. Cells had been harvested after engraftment reached 75% in the peripheral blood or mice health score reached either three at single item or the total score had reached five. T-ALL cells were collected from spleen and cryopreserved as described (Schmitz *et al*, 2011). Blast enrichment in the sample had been evaluated by flow cytometry using same antibody panel. Xenograft identity was verified by DNA fingerprinting using the commercial AmpFlSTR® NGM SElect kit. *In vivo* experiments were approved by the veterinary office of the Canton of Zurich, in compliance with ethical regulations for animal research.

### Whole-exome sequencing

Libraries for whole-exome sequencing were prepared with SureSelectXT Target Enrichment System for Illumina Paired-End Multiplexed Sequencing Library v4/v6 (Agilent, Santa Clara, CA, USA) according to the manufacturer's protocols. DNA concentration was determined with the Qubit fluorometer using BR dsDNA Assay (Qubit 2.0, Invitrogen Life Technologies, Grand Island, NY, USA). 200 ng of genomic DNA was sheared using Covaris S2 instrument (Covaris, Woburn, MA, USA) to a mean size of 150–200 bp. Pooled indexed sample libraries were sequenced in paired-end 100-bp mode using an Illumina HiSeq2000 deep sequencing instrument (Illumina, San Diego, CA, USA). All raw sequencing reads from fastq files were mapped against the human reference genome hg19 [hg19, GRCh37 Genome Reference Consortium Human Reference 37 (GCA_000001405)] using modified variant tools align pipeline bwa_gatk28_hg19 (San Lucas *et al*, 2012). For the PDX samples, we have classified the reads into those that aligned to hg19, mm10, and ambiguous and discarded the reads, which preferentially aligned to mm10. Briefly, alignment was performed with bwa (Burrows-Wheeler Aligner; Li & Durbin, 2009) mem algorithm and followed by removing duplicate reads (Picard) and local realignment by Genome Analysis Toolkit (GATK; McKenna *et al*, 2010). Mutation callers Mutect (do Valle *et al*, 2016) and Strelka (Saunders *et al*, 2012) were next applied on the recalibrated data for detection of somatic mutations. The presence of single nucleotide variants (SNV) and small insertions and deletions (InDels) in the germ line was excluded by analysis of remission samples. Identified variants were functionally annotated using ANNOVAR (Wang *et al*, 2010) tool with the corresponding nucleotide exchange and then compared with those listed in dbSNP v138 (http://www.ncbi.nlm.nih.gov/snp/) and in the 2014 release of the 1000 Genomes Project (http://www.internationalgenome.org/). Annotations also included GERP conservation scores and indications whether the variant is located in a segmental duplicated region (SegDup). Predictions of the functional impact of amino acid exchanges on the structure and function of the respective protein were computed using SIFT and PolyPhen-2. All SNVs were filtered for non-synonymous, stopgain or stoploss variants requiring a frequency of 1% or less in the 1000 Genome Project release 2014 and a minimum of five supporting reads, unless the variant was called in more than one sample from the same patient. Finally, all detected variants were manually curated using the Integrated Genome Viewer.

### Low-coverage whole-genome sequencing

Libraries for low-coverage WGS were prepared using NEBNext Ultra DNA Library Prep Kit for Illumina (New England Biolabs, Frankfurt am Main, Germany) from 100 ng of genomic DNA. Five samples were pooled and sequenced on one Illumina HiSeq 2000 lane generating 150-bp paired-end reads. Mean DNA sequence coverage was threefold (range: two- to fivefold). Adapters were trimmed by cutadapt (Martin, 2011); the sequences were aligned using bwa-mem (Li & Durbin, 2009). Average coverage of 10-kb genomic intervals was calculated using DELLY Cov (Rausch *et al*, 2012). Next, the GC correction was performed, and the normalized read-depth ratios between the leukemic samples and the matched remission samples were calculated and plotted using R packages: DNAcopy, ggplot2, reshape2, and scales.

### DNA methylation analysis using 450k BeadChip Arrays and Infinium® MethylationEPIC BeadChip Arrays

Genomic DNA of 11 out of 12 matched pairs of PDXs and corresponding primary leukemias was available for the analysis and has been isolated or provided by the collaborating centers. DNA concentration was measured by Qubit® 2.0 Fluorometer (Qubit® dsDNA High Sensitivity Assay Kit; Thermo Fischer Scientific). Genomic DNA (200 ng) was bisulfite-converted using the EZ DNA Methylation-Gold Kit (Zymo Research, Irvine, CA, USA). The Infinium® Methylation assay and EPIC assay (Illumina) were carried out as previously described (Bibikova & Fan, 2010). Generated data from the 450k and 850k human methylation arrays were normalized by the Beta MIxture Quantile (BMIQ) method (Teschendorff *et al*, 2013) using the RnBeads analysis software package (Assenov *et al*, 2014). Background correction with ENmix.oob (Xu *et al*, 2016), as well as quality control, preprocessing, exploratory analysis, and differential methylation analysis, was performed using the RnBeads analysis software package (Assenov *et al*, 2014). The methylation level of a CpG locus was expressed as beta value (β), which represents the proportion of methylated alleles divided by all alleles. A gene promoter was defined as the region spanning 1.5 kilobases (Kb) upstream and 0.5 Kb downstream of the respective transcription start site. A threshold of 0.2 in absolute numbers was set to define a difference in methylation level between the initial and relapse samples. Promoters with this 20% difference were considered as an "event". To analyze whether particular promoters were preferentially hypo/hypermethylated in the PDX models, we filtered for the promoters that were represented on the arrays by at least three different probes and had a gene symbol assigned. To define probability of observing a recurrent difference between the primary sample and a PDX model, we calculated a baseline probability of a hypomethylation event (0.003) and a hypermethylation event (0.018). Next, we computed *P*-values for observing a number of events ($k = 1$, 2, 3…22) per gene using a binomial test. Increase in β value (min. 20%) in at least four patients ($P = 0.0278$) and decrease in at least three patients ($P = 0.014$) were statistically significant. Primary leukemias from two of the patients included in this study were part of the analysis reported by us previously (Kunz *et al*, 2015).

### ATAC sequencing

In order to reduce mitochondrial reads in ATAC-Seq libraries, we used a modified ATAC-Seq library preparation protocol, (Corces

### The paper explained

#### Problem

Identifying appropriate preclinical models to study the biology of leukemia *in vivo* and to test novel molecular-targeted therapeutics remains a major challenge. Although patient-derived xenografts (PDXs) of pediatric T-cell leukemia were reported to be generally stable at the genomic level, little is known about conservation of their epigenetic features and chromatin architecture.

#### Results

We performed a detailed multilevel genomic and epigenomic analysis of primary leukemia and matched primary PDX samples collected at the time of initial diagnosis and at relapse from 12 pediatric T-ALL patients.

Patient-derived xenograft models largely preserved genetic alterations, and in many cases, the model truly reflected the leukemia's subclonal composition and clonal hierarchy. Samples of patients who later developed a type 1 relapse (all clonal mutations detected at the time of initial disease maintained at relapse) were observed to be more stable in the PDX mice than those that have been obtained from patients with a type 2 relapse (a subset of clonal mutations detected at initial disease lost at relapse).

The epigenetic profile measured through detecting DNA methylation across the genome was largely preserved in PDX mice in (97.5% of the promoters preserved; $\rho = 0.99$). Likewise, the genome-wide chromatin accessibility profile analyzed by ATAC-Seq was well preserved ($\rho = 0.96$). However, a minor fraction of gene loci associated with immune response and with response to cytokine stimulus show reduced chromatin accessibility (3.7% of the ATAC-Seq peaks) or hypermethylation (0.2% of the promoters) upon propagation in mice. This observation might be a result of the absence of a normal immune system and the reduced ability of murine cytokines and chemokines to replace their human counterparts.

#### Impact

Our results indicate a remarkable genomic stability and for the first time document the preservation of the epigenomic landscape in pediatric T-cell leukemia-derived primary NSG mice.

We conclude that while some of the complexity of the genomic and epigenetic characteristics and the subclonal architecture is likely to be remodeled in the course of engraftment, the primary PDX models represent suitable surrogates for the study of disease biology and for the preclinical development of novel treatment strategies in pediatric T-cell.

*et al*, 2016) which uses digitonin instead of NP40 to selectively permeabilize cell membranes and not the mitochondrial membranes. Fifty thousand cryopreserved PDX and primary cells were used for library preparation. After thawing and washing with complete culture medium (RPMI medium with 10% FBS and 1% penicillin–streptomycin), living cells were enriched for lymphoblasts using density gradient centrifugation. 15 ml Falcon tube was filled with 6 ml ficoll (1.077 g/ml), and 1 ml of defrosted cells was carefully laid on the solution. Centrifugation with no brake for 20 min at 400 *g* was performed to form lymphoblast monolayer. After enrichment, cell number and viability were assessed with trypan blue staining. Enrichment was repeated until the viability exceeded 95% if the number of cells was sufficient for multiple centrifugation steps. DNA libraries prepared with the modified ATAC-Seq protocol were sequenced on Illumina platforms (NextSeq500). Data analysis was carried out using an in-house developed ATAC-Seq pipeline (https://github.com/tobiasrausch/ATACseq).

Briefly, the pipeline first discarded mouse contaminating reads that preferentially aligned to mm10 [PDX: 2.41%, primary samples (PRI): 0.02%]. Reads that aligned to hg19 and ambiguous reads that were not classified as hg19 or mm10 were used for further analysis. Adapters were trimmed by cutadapt (Martin, 2011), and the trimmed sequences were aligned to the human reference genome (hg19) by Bowtie2 (Langmead & Salzberg, 2012). Reads aligning to mitochondrial DNA (PDX: 13.85%, PRI: 9.67%), duplicate reads, reads with mapping quality below 30, unmapped reads, and unplaced contigs (PDX: 25.19%, PRI: 29.21%) were removed using samtools (Li *et al*, 2009). Peak calling was performed using MACS2 (Zhang *et al*, 2008) with user-specified parameters, and the IDR method (Li *et al*, 2011) was used to identify significant peaks above the background noise. Peak annotation with genomic features was done using the Homer package (Heinz *et al*, 2010). For quality control transcription start site enrichment values, the insert size distribution with the characteristic nucleosome pattern, and the fraction of reads in peaks, was used. All quality control metrics were computed using Alfred (https://github.com/tobiasrausch/alfred), which was also used to create browser tracks (bedGraph files). Differential peak calling was performed with DESeq2 (Love *et al*, 2014) in the multi-factor design mode to control for additional variation resulting from a high interpatient heterogeneity with a log-fold change threshold set to 0.1.

### Multiplex ligation-dependent probe amplification (MLPA)

The commercially available SALSA MLPA P383 T-ALL probe mix (MRC-Holland, Amsterdam, The Netherlands) and a custom-made probe set based on the SALSA MLPA P200-A1 probemix (MRC-Holland) were used for the detection of specific copy number variations as described before (Richter-Pechanska *et al*, 2017).

### Software and bioinformatical tools

Graphical representation and statistics were done using: R (R Core Team, 2017), GraphPad Prism version 6.00 for Windows (La Jolla, CA, USA).

Functional enrichment analyses for hyper-/hypo-accessible ATAC regions and their graphical representation were generated using GREAT (Genomic Regions Enrichment of Annotations Tool; McLean *et al*, 2010).

## Data availability

Sequence data have been deposited at the European Genome-phenome Archive (EGA, http://www.ebi.ac.uk/ega/), which is hosted by the EBI, under accession number EGAS00001003248.

**Expanded View** for this article is available online.

## Acknowledgements

This work was supported by Deutsche José Carreras Leukämie Stiftung, Dietmar Hopp Stiftung and by Bundesministerium für Bildung und Forschung [TRANSCALL—part of the ERA-NET on "Translational Cancer Research" (TRANSCAN)]. We would like to thank Reddy O. Bandapalli for his contribution

in early the early phase of the project. We thank all the patients who participated in the study and their families.

## Author contributions

PR-P designed the experiments, performed the bioinformatic analyses, and wrote the manuscript; JBK designed the research, contributed to the interpretation of the results, and wrote the manuscript; BB established the PDX models, contributed to the interpretation of the results, and wrote the manuscript; CKD performed the methylation experiments and wrote the manuscript; TR performed bioinformatic analyses and aided in interpreting the results; BE-U performed the ATAC-Seq experiments and their analysis; YA performed bioinformatic analysis of the methylation results and contributed to their interpretation; VF established the PDX models and provided the data; BM performed work on the PDX models; SMW provided the idea and contributed to the interpretation of the results of ATAC-Seq; MZ provided the patients' clinical data and their analysis; JS performed MLPA experiments and their analysis; MH performed and analyzed Sanger sequencing data; MSt, MSc, GC, GE, KB, RK-S, and CE provided patients samples and data for the analyses; MUM contributed to the analysis of the results and to the writing of the manuscript; JOK contributed to the analysis and interpretation of the results; J-PB contributed to the design of the research and writing of the manuscript, supervised the establishment of the PDX models; and AEK designed the research, supervised the project, and wrote the manuscript. All authors reviewed and contributed to the final manuscript.

## Conflict of interest

The authors declare that they have no conflict of interest.

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
