## [Review Process File · EMBO Molecular Medicine]

PDX Models Recapitulate the Genetic and Epigenetic Landscape of Pediatric T-Cell Leukemia

Paulina Richter-Pechańska, Joachim B. Kunz, Beat Bornhauser, Caroline Von Knebel Doeberitz, Tobias Rausch, Büşra Erarslan-Uysal, Yassen Assenov, Viktoras Frismantas, Blerim Marovca, Sebastian M. Waszak, Martin Zimmermann, Julia Seemann, Margit Happich, Martin Stanulla, Martin Schrappe, Gunnar Cario, Gabriele Escherich, Kseniya Bakharevich, Renate Kirschner-Schwabe, Cornelia Eckert, Martina U. Muckenthaler, Jan O. Korbel, Jean-Pierre Bourquin, Andreas E. Kulozik

Review timeline:

Submission date:	15 June 2018
Editorial Decision:	18 July 2018
Revision received:	14 September 2018
Editorial Decision:	02 October 2018
Revision received:	09 October 2018
Accepted:	11 October 2018

Editor: Lise Roth

Transaction Report:

1st Editorial Decision

18 July 2018

Thank you for the submission of your manuscript to EMBO Molecular Medicine. We have now heard back from the three referees whom we asked to evaluate your manuscript. As you will see from the reports below, they are overall positive and support publication of the article in EMBO Molecular Medicine pending appropriate revisions. They however highlight the need to further strengthen the study through additional experiments and more thorough discussion and clarifications. EMBO Molecular Medicine encourages a single round of revision only and therefore, acceptance or rejection of the manuscript will depend on the completeness of your responses included in the next, final version of the manuscript.

Please also contact us as soon as possible if similar work is published elsewhere. If other work is published, we may not be able to extend the revision period beyond three months.

I look forward to receiving your revised manuscript.

***** Reviewer's comments *****

Referee #1 (Remarks for Author):

In this study, Richter-Pechanska et al. performed a detailed analysis of 12 diagnosis-relapse pairs and their respective xenografted samples. They largely show stability between the primary material and the xenograft using a variety of assays. This is a nice and important study because these xenograft samples are routinely used for preclinical drug evaluation. However, I have a few comments that might improve the manuscript.

1. Although Supplementary Table 1 does contain some information on the 12 T-ALL cases studied, more information should be provided regarding the genetic subtype, i.e. SIL-TAL1, TLX1, TLX3, HOXA etc. This information is now lacking from the manuscript and is also not evaluated in the progression to relapse and/or the xenograft material. For example, if a sample has a SIL-TAL1, is this retained at relapse and during xenograft. If the authors would have transcriptomics data available from these cases, one could also try to deduce that information from these transcriptional signatures.

2. Did the authors have normal control gDNA available for each case to make sure the SNV/Indels that were used for the conservation analysis are truly somatic?

3. I think it might be beneficial for the reader to have an overview for each sample on the conservation aspect using the different technologies applied here. For example, show for each sample all the SNV/Indels and denote for each of them if they were conserved or not. If conserved, the AF in initial sample and xenograft should be presented in that overview.

4. Is there something specific about type 1 vs type 2 relapse, in relation to genetics, genetic subtype of the T-ALL samples? Also and more specifically, P2, P7, P10 and P12 are interesting cases with a type 2 relapse where the xenografts are more variable. Do these samples (P2, P7, P10 and P12) all belong to the same genetic subtype?

5. Could the authors state or clarify the origin of the xenograft material that was used for their analysis. Has this always been spleen? Or has this be different for the different samples?

Referee #2 (Remarks for Author):

Page 6 - Authors looked at SNVs, indels and CNAs; was there a chance to look at chromosomal rearrangements? (STIL-TAL1 deletion, rearrangements of TLX1, TLX3, TAL1, LMO1, etc.?). It would be expected that such chromosomal rearrangements would be stable between patient sample and PDX, and if not possible by the sequencing method would be easy to achieve by karyotyping/FISH analysis/PCR on the samples.

Page 7: The authors write "Finally, two of these models (P6 and P7) did not preserve some alterations found in the primary initial leukemia. These mutations (SNVs and large deletions of chr7&10) were not detected at relapse either, thus indicating that these changes represent passenger mutations."

I am not sure if one can conclude that these mutations are passenger mutations. Is it not just selection of clones?

General comment on the sequence data:

How sure are the authors about the absence of mutations in the samples? For example in P6, there are only few mutations and the STAT5B N642H mutation (a known driver mutation) is only present in the patient sample at diagnosis, but not in the PDX. What was the coverage for this mutation in the PDX sample? How much contamination of mouse DNA was present in the DNA from the PDX sample? Have the authors checked for the presence of this mutation in IGV (visual inspection of the sequence reads) or by PCR/Sanger sequencing? In order to draw solid conclusions here, coverage should be very high in order to exclude false negative results.

Similarly, on page 8 (PDX specific mutations), the authors speculate that mutations that appear in the PDX models and were not detected at diagnosis in the patient samples are actually not newly acquired in the xenografts, but were missed by the sequencing procedure in the patient sample. It is simple to test this, by going back to the DNA of the patient samples and perform PCR and Sanger sequencing or allele specific PCR. Since the allele frequency of the other mutations did not change significantly, the mutation such as the NOTCH1 mutation was likely missed but should be present at

high allele frequency at diagnosis. It would strengthen the work if this could be tested and included in the manuscript.

Referee #3 (Remarks for Author):

Review of: PDX Models Recapitulate the Genetic and Epigenetic Landscape of Pediatric T-Cell Leukemia

General remarks:

In this work, the authors present the results of genomic and epigenomic analysis of primary T-ALL cells collected at initial diagnosis and relapse, as well as matched patient-derived xenografts from both time points. The authors use the genomics data (exome sequencing and low-coverage whole-genome sequencing) to define dynamics of clonal heterogeneity before and after treatment, and the degree to which those dynamics are recapitulated (or in some cases anticipated) by the PDX models. The authors also use microarrays to study DNA methylation, and ATAC-seq to study chromatin accessibility in order to determine the degree to which epigenetic features are recapitulated in the PDX models.

Overall, the findings in this work with regard to genomic features and clonal dynamics are of interest, although they are generally in keeping with previously published work. The DNA methylation data appears to be supportive of the contention that epigenetic variability between tumors is preserved in the xenografts, although more information could be provided with regard to whether the differentially methylated genes are likely to be of oncogenic significance. These remarks also apply to the more numerically limited ATAC-Seq analysis of chromatin accessibility in PDX versus primary samples, although more information is required for the reader to fully assess the quality and interpretability of these datasets.

Specific comments:

Figure 1a shows a high correlation between variants detected in PDX versus primary samples, but the authors chose to exclude one divergent sample prior to calculating the correlation, a statistically unsatisfying solution. Another approach would be to raise the threshold for allelic fraction to 20% for inclusion of SNPs in calculations for this figure, since the outlier data point is driven by a large number of mutations detected at low frequency in the patient but not in the PDX - the non-thresholded figure could be shown in the supplemental.

The analysis of clonal dynamics in the genomic sequencing datasets is generally a strength of the paper. However, I do not agree that one can conclude that mutations present in the initial sample and lost in both the initial PDX and relapse are necessarily "passenger mutations", which implies that they held no selective advantage in the initial oncogenic process. These could be events that were selected for in the progression of the major population prior to diagnosis, but which were disfavored by both treatment and engraftment, leading to selection for an "earlier" divergent subclone. One helpful measure could be to state whether the genes lost in this pattern were significantly less likely to be known recurrently mutated T-ALL cancer genes.

The DNA methylation analysis is limited to analyses of quality and variability, with no remarks on biological significance of the differentially methylated genes. Were differentially methylated genes in PDX vs primary enriched for particular GO categories or sequence features? Are these genes that have been previously noted to be aberrantly methylated or variably expressed in T-ALL? Alternately, are these predominantly genes that are transcriptionally silent in T-ALL, and thus variation in promoter methylation patterns would be unlikely to lead to altered tumor biology? While transcriptional profiling data was not available for samples in this study, published datasets could be used to categorize the differentially methylated genes broadly with respect to whether they are uniformly expressed, variably expressed, or uniformly silent genes in T-ALL

For ATAC-Seq, the analysis is limited to QC and summary statistics, and additional exploration of these data would enhance the impact of the work. ATAC-Seq data can be subject to significant

protocol-dependent variability driven by factors such as the ratio of nuclei to transposase, and the number of PCR cycles performed. This can result in significant batch effects. It would be helpful to know whether all specimens were processed at the same time and with identical conditions (e.g. number of PCR cycles), or whether samples were processed at different times and with all samples from a given patient processed together, potentially creating batch effects that might drive sample clustering in the principal component analysis. Ideally batch data would be presented in a supplemental table. The authors state that they were unable to obtain high-quality RNA from the primary samples due to sample degradation. Since chromatin accessibility (unlike DNA sequence and DNA methylation) is a highly dynamic process, this raises significant concern for the quality of their chromatin accessibility data.

Accordingly, at least some ATAC-Seq tracks should be shown at sample loci as a visual demonstration of data quality and interpretability - the authors could choose loci that show a high degree of differential accessibility between primary and PDX, and / or genes or enhancers of known significance in T-ALL from prior publications. Given the significant difference in fraction of reads within TSS's for the PDX versus primary samples, it may be necessary to adjust track scale accordingly (or to discard reads outside of peaks) in order to facilitate direct visual comparison between these datasets.

It would be informative to separate out promoters vs distal sites (e.g. enhancers) for the measures of reproducibility and variance presented by the authors. It appears that only just over 10,000 peaks were reproducibly detected in these samples, which is a low number for ATAC-Seq, comparable to the total number of active promoters expected in a cell. Since active enhancer sites are typically many multiples of this, it seems likely that the authors are only detecting the very strongest promoters and enhancers in these datasets. It would also be helpful to compare the degree to which their ATAC-Seq peaks overlap active promoters and enhancers detected in previously published chromatin datasets for T-ALL (e.g. ATAC-Seq or histone methylation / acetylation datasets). Supplemental table 4, which lists detected ATAC-Seq peaks, should denote whether each peak is a proximal or distal site, and the identity of the closest gene (or all linked genes used for GREAT analysis).

The breakdown of promoters vs enhancers has important implications for the authors' claim that immune-related sites are selectively inactivated in the PDX models. Enhancers are generally most abundant near tissue-specific genes, so if the differentially accessible peaks are mostly enhancers, while the unchanged peaks are predominantly promoters, this alone could drive enrichment for GO terms related to immune cells, T cell activation, etc, in a T cell-phenotype population. Performing separate comparisons of differentially accessible promoters and differentially accessible enhancers could help address this concern. Showing example loci with differentially regulated peaks near genes known to be oncogenically significant and / or cytokine-regulated could be helpful. Overall more support is required for the claim of systematically silenced immune signaling-related genes in the PDX models, given that this claim is prominently featured in the abstract and discussion.

1st Revision - authors' response

14 September 2018

***** Reviewer's comments *****

Referee #1 (Remarks for Author):

In this study, Richter-Pechanska et al. performed a detailed analysis of 12 diagnosis-relapse pairs and their respective xenografted samples. They largely show stability between the primary material and the xenograft using a variety of assays. This is a nice and important study because these xenograft samples are routinely used for preclinical drug evaluation. However, I have a few comments that might improve the manuscript.

1. Although supplementary table 1 does contain some information on the 12 T-ALL cases studied, more information should be provided regarding the genetic subtype, ie. SIL-TAL1, TLX1, TLX3, HOXA etc. This information is now lacking from the manuscript and is also not evaluated in the progression to relapse and/or the xenograft material. For example, if a sample has a SIL-TAL1, is this retained at

relapse and during xenograft. If the authors would have transcriptomics data available from these cases, one could also try to deduct that information from these transcriptional signatures.

We agree with the reviewer that information about the leukemia driving events and their preservation between primary samples and corresponding PDX models will be useful. We have thus included this information based on transcriptomic and genomic data of the PDX samples in **Fig. 1c** and in the text of the manuscript (**line 117**):

On the basis of characteristic expression profiles of the samples propagated in PDX we classified patients into the following subgroups: TAL1/2 (n = 5), TLX1/3 (n = 3), HOXA (n = 1), NKX2-4/5 (n = 2) and LMO2 (n = 1; Fig. 1c). In addition, we confirmed the breakpoints of the fusions in the genomic DNA in 6 of 12 primary leukemia samples either by Sanger sequencing (Appendix Figure S2; 3 patients) or by MLPA analysis for SIL-TAL1 fusions (Table EV2a; 3 patients). In all of these cases the driving event was identified in the primary samples obtained at initial disease and relapse, and also in the corresponding PDX sample.

2. did the authors have normal control gDNA available for each case to make sure the SNV/Indels that were used for the conservation analysis are truly somatic

Yes, for each patient a remission sample was available as a control. For the mutation calling we used two independent callers – Strelka and Mutect which use the normal control (remission sample) to detect the somatic variants. After viewing of the alignments and curating the mutations in IGV comparing the leukemia and the remission samples we have created a consensus list of variants for each of the samples. We have now updated the Materials and Methods section to explain our strategy to exclude germ line mutations more explicitly (**line 431**):

Mutation callers Mutect (do Valle et al, 2016) and Strelka (Saunders et al, 2012) were next applied on the recalibrated data for detection of somatic mutations. The presence of single nucleotide variants (SNV) and small insertions and deletions (InDels) in the germ line were excluded by the analysis of remission samples.

3. I think it might be beneficial for the reader to have an overview for each sample on the conservation aspect using the different technologies applied here. For example, show for each sample all the SNV/Indels and denote for each of them if they were conserved or not. If conserved, the AF in initial sample and xenograft should be presented in that overview.

This reviewer may have missed **Table EV2b** of the original manuscript, which contains all detected mutations (preserved/not preserved) with corresponding allele frequencies. We have now emphasized it in the text of the manuscript (**line 126**):

72% (512) of the total 712 SNVs and InDels that were detected in patients' samples were also detected in the corresponding PDXs (all SNVs/InDels with corresponding AF are listed in **Table EV2b**).

In addition, we have now added **Table EV3**, which contains the fraction of preserved mutations and number of differences in the methylation and chromatin accessibility profiles between the primary leukemia and the corresponding PDX per sample:

- A) (#)% of preserved clonal (AF \geq 30%) mutations
- B) (#)% of preserved subclonal (AF:10-30%) mutations
- C) (#)% of preserved subclonal (AF<30%) mutations
- D) (#)% of preserved CNA (lcWGS)/ total number of CNA detected in the primary sample
- E) (#)% of preserved del/amp (MLPA)/ total number of del/amp detected in the primary sample
- F) (#)% of promoters hypomethylated upon engraftment ($\beta \geq 0.2$) out of 21,711
- G) (#)% of promoters hypermethylated upon engraftment ($\beta \geq 0.2$)
- H) (#)% of peaks at least 2-fold down in PDX
- I) (#)% of peaks at least 2-fold up in PDX

This overview was then referred to in the main text of the manuscript (**line 151**):

An overview of the fraction of preserved mutations for each of the samples is listed in **Table EV3**.

and (line 234):

The number of differences detected at the epigenetic level between primary leukemia and the matched PDX for each of the analyzed samples is listed in Table EV3.

4. Is there something specific about type 1 vs type 2 relapse, in relation to genetics, genetic subtype of the T-ALL samples? Also and more specifically, P2, P7, P10 and P12 are interesting cases with a type 2 relapse where the xenografts are more variable. Do these samples (P2, P7, P10 and P12) all belong to the same genetic subtype?

TAL1 was recognized as a driver predominantly in type 2 in 4 of 7 leukemias (among these are the three mentioned by the reviewer: P2, P10, P12; the driver of P7 is TLX3, see Fig 1c). By contrast, we did not identify TAL1 fusions in type 1 leukemias. However, this trend is not significant ($p=0.20$, Fisher's exact test). Possibly due to the small number of samples, we did not identify significant differences between leukemias that relapsed as either type 1 or type 2.

5. Could the authors state or clarify the origin of the xenograft material that was used for their analysis. Has this always been spleen? Or has this be different for the different samples?

We always used spleen samples for the analyses reported here. As suggested, we have now explicitly stated this in (line 409):

T-ALL cells were collected from spleen and cryopreserved as described (Schmitz et al, 2011).

Referee #2 (Remarks for Author):

Page 6 - Authors looked at SNVs, indels and CNAs; was there a chance to look at chromosomal rearrangements? (STIL-TAL1 deletion, rearrangements of TLX1, TLX3, TAL1, LMO1, etc.). It would be expected that such chromosomal rearrangements would be stable between patient sample and PDX, and if not possible by the sequencing method would be easy to achieve by karyotyping/FISH analysis/PCR on the samples.

As suggested and as detailed in response to comment 1 of reviewer #1, we have now included new data showing the chromosomal rearrangements known to drive T-ALL leukemogenesis and that these are stable in primary and PDX samples.

Page 7: The authors write "Finally, two of these models (P6 and P7) did not preserve some alterations found in the primary initial leukemia. These mutations (SNVs and large deletions of chr7&10) were not detected at relapse either, thus indicating that these changes represent passenger mutations."

I am not sure if one can conclude that these mutations are passenger mutations. Is it not just selection of clones?

We are grateful to this reviewer and also to reviewer #3 to raise this alternative interpretation. Indeed, clonal selection is an alternative mechanism that may explain this finding. We have thus modified this sentence accordingly (line 176):

Finally, two of these models (P6 and P7) did not preserve some alterations (SNVs and large deletions of chr7&10) found in the primary initial leukemia. These mutations were not detected at relapse either, which may indicate that these mutations represent passenger mutations. Alternatively, these mutations may have been present in subclones that were selected for in the progression of the major population prior to diagnosis, but were disfavored by both treatment and engraftment, leading to selection for an "earlier" divergent subclone.

General comment on the sequence data:

How sure are the authors about the absence of mutations in the samples? For example in P6, there are only few mutations and the STAT5B N642H mutation (a known driver mutation) is only present in the patient sample at diagnosis, but not in the PDX. What was the coverage for this mutation in

the PDX sample? How much contamination of mouse DNA was present in the DNA from the PDX sample? Have the authors checked for the presence of this mutation in IGV (visual inspection of the sequence reads) or by PCR/Sanger sequencing? In order to draw solid conclusions here, coverage should be very high in order to exclude false negative results.

Similarly, on page 8 (PDX specific mutations), the authors speculate that mutations that appear in the PDX models and were not detected at diagnosis in the patient samples are actually not newly acquired in the xenografts, but were missed by the sequencing procedure in the patient sample. It is simple to test this, by going back to the DNA of the patient samples and perform PCR and Sanger sequencing or allele specific PCR. Since the allele frequency of the other mutations did not change significantly, the mutation such as the NOTCH1 mutation was likely missed but should be present at high allele frequency at diagnosis. It would strengthen the work if this could be tested and included in the manuscript.

For the mutation calling we used two independent callers – Strelka and Mutect which use the normal control (remission sample) to detect the somatic variants. We set the threshold of detection at 10% unless a mutation was found in more than one sample from a set of samples derived of one patient. We have created the consensus from the output of the variant callers and we manually curated all of them in IGV. As suggested, we have now added the following sentence (line 444):

Finally, all detected variants were manually curated using the Integrated Genome Viewer.

In this manner we could detect mutations such as STAT5B N642H in P2 in only 4 of 71 reads in the corresponding PDX sample and only one supporting read for STAT5B mutation in the PDX of model P6 (1/86). In contrast for mutations lost in the PDX sample derived of P12 relapse we did not find any reads supporting mutations despite a very high coverage up to 135 reads. Following the suggestions of the referee we have included the read depth information for all of the detected and discrepant clonal mutations in **the Table EV2b**.

Further, we have now performed the suggested independent analyses to validate these results by Sanger sequencing of the PDX samples in order to minimize the possibility of false positive results. We sequenced STAT5B N642H in P6INI; DNMT2 (R199Q), FOXO3 (R211Q) in P7INI, PIK3CA (R1023Q), and MAP1B (T908K) in P12REL. We could confirm the results generated by WES and the absence of the mutations in the PDX models. The results are now depicted in the **Appendix Figure S5a** and mentioned in the text of the manuscript (line 190):

The absence of the variants in the matched PDXs models was confirmed by Sanger sequencing for a subset of 5 mutations in *STAT5B* (N642H), *DNMT2* (R199Q), *FOXO3* (R211Q), *PIK3CA* (R1023Q), and *MAP1B* (T908K; Appendix Figure S5a).

We agree with this reviewer that murine contamination is a potential source of error in the analysis of PDX samples. However, the hybridization based library preparation protocol (SureSelect, Agilent) ensured that murine contamination in the WES is not substantial. During hybridization of the probes designed to capture human sequences, the majority of the murine DNA fragments is washed out and the human sequences are enriched. Moreover, we have performed alignment to both hg19 and mm10 and discarded all the reads which preferentially aligned to the murine genome. The remaining reads were either preferentially binding to hg19 or represented ambiguous reads. As the ambiguous reads were only a minor fraction (1-2%) of all the reads, they do not have a relevant impact on the AF of detected mutations. We have now explicitly explained this strategy of minimizing the contamination of the data by murine reads (line 426):

For the PDX samples we have classified the reads into those that aligned to hg19, mm10 and ambiguous and discarded the reads, which preferentially aligned to mm10.

As suggested, we have also performed Sanger sequencing for a subset of mutations that were detected only in the PDX samples but not in the matched primary leukemias (NOTCH1, SGCE, WDR88, MAST3, PCDH15 and COL6A3). The results of Sanger sequencing confirmed the absence of the mutations in the primary samples. We now show these new data in **Appendix Figure S5b** and have modified the text of the manuscript accordingly (line 206):

The absence of mutations in *NOTCH1*, *SGCE*, *WDR88*, *MAST3*, *PCDH15* and *COL6A3* in the primary leukemias (P1INI, P5INI, P10INI, P10REL and P12REL) was validated by Sanger sequencing as shown in Appendix Figure S5b.

Referee #3 (Remarks for Author):

Review of: PDX Models Recapitulate the Genetic and Epigenetic Landscape of Pediatric T-Cell Leukemia

General remarks:

In this work, the authors present the results of genomic and epigenomic analysis of primary T-ALL cells collected at initial diagnosis and relapse, as well as matched patient-derived xenografts from both time points. The authors use the genomics data (exome sequencing and low-coverage whole-genome sequencing) to define dynamics of clonal heterogeneity before and after treatment, and the degree to which those dynamics are recapitulated (or in some cases anticipated) by the PDX models. The authors also use microarrays to study DNA methylation, and ATAC-seq to study chromatin accessibility in order to determine the degree to which epigenetic features are recapitulated in the PDX models.

Overall, the findings in this work with regard to genomic features and clonal dynamics are of interest, although they are generally in keeping with previously published work. The DNA methylation data appears to be supportive of the contention that epigenetic variability between tumors is preserved in the xenografts, although more information could be provided with regard to whether the differentially methylated genes are likely to be of oncogenic significance. These remarks also apply to the more numerically limited ATAC-Seq analysis of chromatin accessibility in PDX versus primary samples, although more information is required for the reader to fully assess the quality and interpretability of these datasets.

Specific comments:

Figure 1a shows a high correlation between variants detected in PDX versus primary samples, but the authors chose to exclude one divergent sample prior to calculating the correlation, a statistically unsatisfying solution. Another approach would be to raise the threshold for allelic fraction to 20% for inclusion of SNPs in calculations for this figure, since the outlier data point is driven by a large number of mutations detected at low frequency in the patient but not in the PDX - the non-thresholded figure could be shown in the supplemental.

As suggested, we have changed the threshold of the allele frequency to 20% in the main Fig. 1a and did not exclude the divergent sample, but calculated the correlation for all of the samples ($R^2=0.882$). The figure with previous allele frequency threshold of 10% was moved to the supplement (Appendix Figure S3). We have introduced the changes in text of the manuscript as follows (line 131):

The proportion of SNVs preserved was higher at initial disease (175/213; 82%) compared to relapse (337/499; 68%), although this difference was mainly driven by a single relapse sample from patient P1 where one subclone with 98 mutations with an allele frequency below 20% did not engraft (Appendix Figure S3).

The analysis of clonal dynamics in the genomic sequencing datasets is generally strength of the paper. However, I do not agree that one can conclude that mutations present in the initial sample and lost in both the initial PDX and relapse are necessarily "passenger mutations", which implies that they held no selective advantage in the initial oncogenic process. These could be events that were selected for in the progression of the major population prior to diagnosis, but which were disfavored by both treatment and engraftment, leading to selection for an "earlier" divergent subclone. One helpful measure could be to state whether the genes lost in this pattern were significantly less likely to be known recurrently mutated T-ALL cancer genes.

We are grateful to this reviewer and also to reviewer #2 to raise this alternative interpretation. We

have now explicitly mentioned the alternative explanation of clonal selection as a possible mechanism for the loss of mutations in the PDX samples as suggested (line 176):

Finally, two of these models (P6 and P7) did not preserve some alterations (SNVs and large deletions of chr7&10) found in the primary initial leukemia. These mutations were not detected at relapse either, which may indicate that these mutations represent passenger mutations. Alternatively, these mutations may have been present in subclones that were selected for in the progression of the major population prior to diagnosis, but were disfavored by both treatment and engraftment, leading to selection for an "earlier" divergent subclone.

The DNA methylation analysis is limited to analyses of quality and variability, with no remarks on biological significance of the differentially methylated genes. Were differentially methylated genes in PDX vs primary enriched for particular GO categories or sequence features? Are these genes that have been previously noted to be aberrantly methylated or variably expressed in T-ALL?

Alternately, are these predominantly genes that are transcriptionally silent in T-ALL, and thus variation in promoter methylation patterns would be unlikely to lead to altered tumor biology?

While transcriptional profiling data was not available for samples in this study, published datasets could be used to categorize the differentially methylated genes broadly with respect to whether they are uniformly expressed, variably expressed, or uniformly silent genes in T-ALL.

We are grateful for this helpful and constructive suggestion. We have correlated methylation patterns with expression data in publicly available data sets of 264 T-ALL patients (Liu et al, The genomic landscape of pediatric and young adult T-lineage acute lymphoblastic leukemia, *Nature Genetics* volume 49, pages 1211–1218 (2017)).

As suggested we distinguished three categories of genes: i) uniformly expressed (mean FPKM>5) ii) variable ($SD>3$) and iii) uniformly silenced (remaining) and analyzed the subset of genes hypermethylated upon engraftment (483) in the context of the entire expression profile. Furthermore we have performed functional enrichment analysis of the genes, whose expression might be affected by hypermethylation upon propagation in mice.

We have now summarized this analysis in the text of the manuscript as follows (line 237):

As the 69 promoters in which we observe reduced methylation levels in PDX are most likely a result of the low blast content in some of the primary samples (Fig. EV2) we focused on the 484 promoters hypermethylated in the PDXs models. Publicly available expression data from 264 T-ALL patients (Data ref: Liu et al, 2017) were available for 408 of these genes. Of the 408 genes only 12 (3 %) had an average expression of at least 5 FPKM and additional 21 genes (5%) had a variable expression ($SD\geq 3$, expressed with at least 5 FPKM in at least 3 patients). In comparison, 43% (8,412/19,464) of the entire transcriptome were categorized as either expressed or variably expressed. 375 of the 408 genes hypermethylated in PDX were not expressed at all or expressed at a very low level (average FPKM ≤ 2), indicating that the vast majority of hypermethylated genes are transcriptionally silent in T-ALL (Fig. EV3).

Functional enrichment analysis of the remaining 33 genes (listed in Table EV4) that were (variably) expressed yielded a term GO:0060761: "negative regulation of response to cytokine stimulus" as most significantly enriched ($FDR=7.33e-03$). Altogether these data suggest that the majority of the genes (92%) that were hypermethylated in PDX were already silenced prior to the propagation in mice, which is in accordance with previously published observations that methylation might serve not to actually repress previously transcribed genes, but maintains their continued silencing in somatic tissues that might otherwise be permissive to these genes (Klutstein et al, 2016). In a small proportion of differentially methylated promoters (8%; 0.17% of all the promoters) the increase in the methylation level in PDX models might be a result of the incomplete interaction between murine cytokines and human receptors.

For ATAC-Seq, the analysis is limited to QC and summary statistics, and additional exploration of these data would enhance the impact of the work. ATAC-Seq data can be subject to significant protocol-dependent variability driven by factors such as the ratio of nuclei to transposase, and the number of PCR cycles performed. This can result in significant batch effects. It would be helpful to know whether all specimens were processed at the same time and with identical conditions (e.g. number of PCR cycles), or whether samples were processed at different times and with all samples

from a given patient processed together, potentially creating batch effects that might drive sample clustering in the principal component analysis. Ideally batch data would be presented in a supplemental table.

As suggested, we have now included a table (**Table EV5a, referred to in line 269**), which contains the technical aspects of the ATAC-seq analysis. We have included the data on the batches (date of the library prep, date of the sequencing/sequencing lane, lot number of the enzyme used, number of PCR cycles, viability of the cells subjected to the ATAC-Seq library preparation and the Ficoll centrifugations details) as well as the quality metrics of the libraries (such as: duplicate fraction, error rate, the number of filtered peaks, fraction of mitochondrial chromosome, peak saturation, fraction mapped to the same chromosome, TSS enrichment, number of unfiltered peaks, unmapped fraction). The quality metrics are well within the acceptable range of current standards published by ENCODE. The lower TSS enrichment score observed in primary samples likely reflects the pre-analytical effects affecting primary samples that have been collected at multiple study centers which, in turn, affected the generated library quality. We processed samples derived from one patient in different batches thus ensuring that potential batch effects do not significantly influence the data quality and to drive the clustering of samples in the PCA.

The authors state that they were unable to obtain high-quality RNA from the primary samples due to sample degradation. Since chromatin accessibility (unlike DNA sequence and DNA methylation) is a highly dynamic process, this raises significant concern for the quality of their chromatin accessibility data.

Bulk RNA-Seq requires app. 500 ng of good quality total RNA, which corresponds to several million T-ALL cells. By contrast, ATAC-Seq requires only app. 50,000 cells. The number of cells was not sufficient to extract the required quantity of RNA for the generation of RNA-Seq libraries from the primary leukemias. Moreover, although there is no systematic analysis comparing directly RNA-Seq and ATAC-Seq performance, it was shown in different applications that the RNA is the most labile analyte (Hewitt et. al. The Impact of Pre-analytic Factors In The Design and Application of Integral Biomarkers for Directing Patient Therapy, Clin Cancer Res. 2012 Mar 15; 18(6): 1524–1530). We may therefore expect that DNA-based ATAC-Seq libraries are more robust than the RNA-Seq libraries. Regarding the concerns about the quality of the ATAC-Seq libraries from the primary samples we have now performed in-depth QC check (**Table EV5a**). As expected, this analysis indicates that the technical quality of the primary material is indeed not quite as high as the quality of the material obtained from PDX models, although well within the acceptable range (**Table EV5a, Fig. EV4**).

Accordingly, at least some ATAC-Seq tracks should be shown at sample loci as a visual demonstration of data quality and interpretability - the authors could choose loci that show a high degree of differential accessibility between primary and PDX, and / or genes or enhancers of known significance in T-ALL from prior publications. Given the significant difference in fraction of reads within TSS's for the PDX versus primary samples, it may be necessary to adjust track scale accordingly (or to discard reads outside of peaks) in order to facilitate direct visual comparison between these datasets.

As suggested, we now show representative ATAC seq tracks to exemplify the quality of the ATAC-Seq data and their preservation between the primary leukemias and the PDX models. These tracks include known leukemia drivers (such as TLX3, NOTCH1 and IL7R) and are shown in **Appendix Figure S7 panel A**. As suggested, we have also selected the most differentially accessible peaks present in either PDX or in primary samples only and presented them in **Appendix Figure S7 in panel B) and C)** respectively (**line 297**).

Representative ATAC-Seq tracks for known leukemia drivers that are accessible both, in the primary and in the PDX samples and for the differentially accessible peaks are shown in Appendix Figure S7.

It would be informative to separate out promoters vs distal sites (e.g. enhancers) for the measures of reproducibility and variance presented by the authors. It appears that only just over 10,000 peaks were reproducibly detected in these samples, which is a low number for ATAC-Seq, comparable to the total number of active promoters expected in a cell. Since active enhancer sites are typically

many multiples of this, it seems likely that the authors are only detecting the very strongest promoters and enhancers in these datasets. It would also be helpful to compare the degree to which their ATAC-Seq peaks overlap active promoters and enhancers detected in previously published chromatin datasets for T-ALL (e.g. ATAC-Seq or histone methylation / acetylation datasets). Supplemental table 4, which lists detected ATAC-Seq peaks, should denote whether each peak is a proximal or distal site, and the identity of the closest gene (or all linked genes used for GREAT analysis).

We have now addressed this important point by splitting the analysis into transcription start site (TSS) regions (± 1000 bp from the promoter) and non-TSS distal regions.

Further, we have to clarify a misunderstanding. In Suppl. Tab. 4 of the original version of the manuscript we have presented the matrix of TSS regions as this seemed to be the easiest way to visualize the genes of interest. To address the fully justified concerns of this referee, we now complemented the table with all the non-TSS regions (indicated as nonTSS and TSS in the table). The much higher number of ATAC-peaks is now shown in **Table EV5a**. We have performed a comparison of the number of peaks with publicly available dataset of characterized AML and sorted healthy hematopoietic cells (Corces et. al., Lineage-specific and single cell chromatin accessibility charts human hematopoiesis and leukemia evolution, *Nature Genetics* (2016)) and found that the range of filtered peaks (15,000-42,000) is similar to the publicly available data analyzed with our pipeline (see boxplot below). A total of 77,344 peaks (TSS and non-TSS) were found in the 15 samples (6 pairs, + 3 biological replicates for PDX samples). As suggested, we have annotated the peaks additionally with the nearest genes as shown in **Table EV5b**.

As there are no published ATAC-Seq datasets on T-ALL, it was not possible to perform a direct comparison. We have therefore compared our ATAC-Seq data with the previously published ChIP-Seq data for the T-ALL cell line DND-41 and summarized this analysis in **Fig. EV5** and the text of the manuscript (**line 286**):

Moreover, a comparison with previously published chromatin datasets for the T-ALL cell line DND-41 (Knoechel et al, 2014) in which we computed expected values based on the randomly shuffled peaks, shows a high degree of overlap between the ATAC-peaks and the active promoters and enhancers detected in histone methylation/acetylation analysis by chromatin immunoprecipitation and sequencing (Fig. EV5).

The breakdown of promoters vs enhancers has important implications for the authors' claim that immune-related sites are selectively inactivated in the PDX models. Enhancers are generally most abundant near tissue-specific genes, so if the differentially accessible peaks are mostly enhancers, while the unchanged peaks are predominantly promoters, this alone could drive enrichment for GO terms related to immune cells, T cell activation, etc, in a T cell-phenotype population. Performing separate comparisons of differentially accessible promoters and differentially accessible enhancers could help address this concern. Showing example loci with differentially regulated peaks near genes known to be oncogenically significant and / or cytokine-regulated could be helpful. Overall more support is required for the claim of systematically silenced immune signaling-related genes in the PDX models, given that this claim is prominently featured in the abstract and discussion.

As suggested, we have now repeated the analysis of peaks differentially accessible in primary and in PDX samples using DESeq2 ($p < 0.05$, lfc threshold=0) after separating the promoter and the enhancer/repressor regions.. We have performed the analysis for all the peaks (77,344) and separately for the TSS-peaks (10,755) and non-TSS distal peaks (66,589). The numbers of differentially accessible promoters are shown in the table below:

	all	TSS	nonTSS
# of peaks	77,344	10,755	66,589
DOWN in PDX	2,887(3.7%)	321(2.9%)	2,369(3.6%)
UP in PDX	2,667(3.4%)	275(2.5%)	1,679(2.5%)

Functional enrichment analysis of the peaks, which were more accessible in PDX models did not yield any significantly enriched functional terms, which suggests that the higher accessibility of some of the peaks in the PDX models is caused by the higher technical quality of the PDX samples, which were treated uniformly, whereas the primary leukemia samples were subjected to different pre-analytical factors. By contrast, peaks that were less accessible in PDXs were highly enriched in terms related to immune system and cytokine production. This enrichment could also be observed in both, the TSS and the non-TSS regions.

We have modified the text of the manuscript accordingly (line 293):

Out of 77,344 (10,755 TSS and 66,589 non-TSS) peaks called in the analyzed sample pairs, 2,667(3.4%) showed significantly increased accessibility (hyper-accessible) and 2,887(3.7%) had significantly decreased accessibility (hypo-accessible) in PDX samples in comparison to the primaries (p<0.05; Fig. 4e; Table EV5b). Representative ATAC-Seq tracks for known leukemia drivers that are accessible both, in the primary and in the PDX samples and for the differentially accessible peaks are shown in Appendix Figure S7.

Functional enrichment analysis was performed separately for the TSS peaks and for the distal (non-TSS) peaks. In both cases, this analysis did not yield any ontology term within the hyper-accessible peaks, whereas the hypo-accessible signature was highly enriched in terms associated with immune/defense responses, cytokine production and leukocyte activation (Fig. 4f). These data suggest a reduced interaction between the immune system and the leukemia in PDX models and possibly the incapability of murine cytokines to stimulate corresponding responses. As a result, in PDX models we observed induced chromatin condensation of the gene regulatory elements involved in immune function and in the regulation of cytokine production of in comparison to their matched primary leukemias (examples are shown in the Appendix Figure S7).

Due to the fact that this enrichment was observed in both methylation and in ATAC-Seq analyses we have opted to maintain the claim of a systematic silencing of immune signaling related genes. However due to the interplay of numerous factors related to the quality of the samples and in the absence of expression data in the primary samples we have toned this claim down. We have thus modified the abstract (line 62):

Interestingly, both the ATAC-regions, which showed a significant decrease in accessibility in PDXs and the regions hypermethylated in PDXs, were associated with immune response, which might reflect the immune-deficiency of the mice and potentially the incomplete interaction between murine cytokines and human receptors.

and the discussion to now read (line 327):

Interestingly, however, a 3.7% fraction of peaks with significantly decreased accessibility in PDX mice as well as 33 promoters recurrently hypermethylated in PDXs when compared to the primary samples are highly enriched for immune function categories, which suggests that the immunodeficient background of the mice has a noticeable effect on the chromatin landscape and methylation patterns of the leukemia cells.

Thank you for the submission of your revised manuscript to EMBO Molecular Medicine. We have now received the enclosed report from the 3 referees that were asked to re-assess it. As you will see the reviewers are now supportive, and I am pleased to inform you that we will be able to accept your manuscript pending minor editorial amendments.

***** Reviewer's comments *****

Referee #1 (Remarks for Author):

All comments have been addressed.

Referee #2 (Remarks for Author):

The authors have addressed all my concerns. The manuscript is now much improved. I have no further questions/remarks.

Referee #3 (Remarks for Author):

The authors have addressed all of my significant concerns in their rebuttal and revised manuscript. This is fine work that would make a substantive contribution to the literature.

Corresponding Author Name: Andreas Kulozik
Journal Submitted to: EMBO Molecular Medicine
Manuscript Number: EMM-2018-09443